# Polyene Antibiotics Physical Chemistry and Their Effect on Lipid Membranes; Impacting Biological Processes and Medical Applications

**DOI:** 10.3390/membranes12070681

**Published:** 2022-06-30

**Authors:** Tammy Haro-Reyes, Lucero Díaz-Peralta, Arturo Galván-Hernández, Anahi Rodríguez-López, Lourdes Rodríguez-Fragoso, Iván Ortega-Blake

**Affiliations:** 1Instituto de Ciencias Físicas, Universidad Nacional Autónoma de México, Av. Universidad s/n, Col. Chamilpa, Cuernavaca 62210, Morelos, Mexico; tammyharo@icf.unam.mx (T.H.-R.); lucero@icf.unam.mx (L.D.-P.); arturo@icf.unam.mx (A.G.-H.); 2Facultad de Farmacia, Universidad Autónoma del Estado de Morelos, Cuernavaca 62210, Morelos, Mexico; yaneth.rodriguez@uaem.edu.mx (A.R.-L.); mrodriguezf@uaem.mx (L.R.-F.)

**Keywords:** polyenes, chemical properties, mechanism of action, semi-synthetic derivatives, liposomal formulation, membrane structure, clinical applications

## Abstract

This review examined a collection of studies regarding the molecular properties of some polyene antibiotic molecules as well as their properties in solution and in particular environmental conditions. We also looked into the proposed mechanism of action of polyenes, where membrane properties play a crucial role. Given the interest in polyene antibiotics as therapeutic agents, we looked into alternative ways of reducing their collateral toxicity, including semi-synthesis of derivatives and new formulations. We follow with studies on the role of membrane structure and, finally, recent developments regarding the most important clinical applications of these compounds.

## 1. Introduction

There are many examples of the involvement of cell membranes in physiological processes. It is now clear that this molecular self-assembled structure is far more complex than Singer and Nicholson’s original idea [1]. It is an ordered soft material with a very rich composition which presents multiple phases simultaneously. This richness plays a role in many processes occurring across, in or through the membrane.

The phenomenon of the interaction of polyenes with the lipid membrane has been studied for decades. Polyenes are quite small molecules, for biological standards, but present a wide variety of phenomena depending on concentration [2,3], solvent [4,5,6,7], pH [8,9,10,11], oxidation [12,13,14,15,16], and membrane properties [17,18,19,20]. Pertinent studies have advanced our understanding of the interaction between molecules and membranes. We reviewed some of these works and added some results of our own. Additionally, polyenes are very important drugs for the treatment of mycosis, parasitosis, and other illnesses, which has increased interest in these compounds. We therefore also included an update on their therapeutic use.

First, we address some polyene molecules and their chemical properties such as structure, solvation, aggregation, oxidation, etc. Then, we review the membrane-based mechanisms of action of the pharmacological activity. We also address examples of successful semi-synthetic derivatives, lipid-based formulations for the delivery of polyenes, and the complexity of membrane structure in relation to polyene action. We finally address recent developments of the clinical use of polyenes.

Some examples of polyene antimycotics are filipin, amphotericin B, nystatin, and natamycin. The latter three are in the World Health Organization’s 22nd list of essential medicines for 2021 [21]. Polyene antibiotics are potent antifungal agents currently used in human therapy. Amphotericin B (AmB) 1 was isolated from *Streptomyces nodosus* in 1955 [3,22] and became commercially available in 1959 [23]. It is still considered the gold standard for the treatment of serious invasive fungal infections [24]. The fermentation by bacteria produces a compound of three forms of amphotericin (A, B, C), which leads to purity problems, mainly in semi-synthetic derivatives. Recently, a genetically engineered strain enhanced production of AmB exclusively, which will lead to purer forms of the polyene and its derivatives [25]. Furthermore, a recent review addressed many other forms of polyenes discovered through genome mining. These discoveries could lead to development of better antifungals [26].

AmB is also employed to treat infections caused by parasitic protozoa, for example leishmaniasis [27,28], as well as prions and viruses [29,30,31,32,33], but the large collateral toxicity of the polyene has prevented a more extensive use [34]. There have therefore been many attempts to reduce this toxicity via derivatives or formulations. The most common way of administering AmB is intravenously in a solution containing deoxycholate to improve AmB solubility in water [35]. Nystatin (Nys) 2 was the first macrolide polyene antimycotic to be discovered [22]; it was first called fungicidin [36] and is synthesized by *Streptomyces noursei*. Due to a slightly higher toxicity, Nys is used topically [37,38,39,40] to treat, for example, oral candidiasis [41]. Toxicity is a major problem when it comes to the clinical use of polyenes and remains an obstacle when addressing an increasing number of multidrug-resistant fungal pathogens such as *Candida albicans*, *Candida lusitaniae* [42], and *Candida auris* [43]. In addition to their function as therapeutic agents, polyenes have long been studied to understand all the phenomena they display at a molecular level and their effect on pharmacological action [44,45,46,47,48,49,50,51,52]. A better understanding of the mechanisms of action involved in the antifungal activity of polyenes and its toxic side effects is needed for the discovery and design of equally potent yet safe alternatives [53]. One major player in the action of polyene antimycotics is the cell membrane. A particular component of the membrane, the sterol, is crucial and, possibly, directly involved in the formation of pores [54,55,56,57] as well as indirectly via modulation of the membrane structure [58,59], or even as the direct target of the drug [60,61].

## 2. Polyene Structure and Chemical Behavior

### 2.1. Chemical Structure of Polyenes

The structure of polyene antibiotics consists of a macrolactone ring of polyunsaturated carbons, where the hydroxyl groups confer the amphipathic character of the molecule. These drugs contain a set of four to eight conjugated double bonds within the macrolactone ring (Figure 1) [62]. The hydrophobic chain is believed to be important in one of the proposed modes of action; it is considered to be responsible for the interaction of antimycotics with sterols for the formation of transmembrane pores [11,54].

Depending on the number of conjugated double bonds, polyenes can be classified into trienes, tetraenes, pentaenes, hexaenes, and heptaenes [2]. These differences are responsible for distinct behavior among the polyenes. For instance, Nys is fluorescent because it contains a tetraene chromophore and a diene; the fluorescence present in this molecule is used to examine its state of aggregation and its interaction with phospholipid bilayer membranes [63]. The fluorescence intensity of Nys in methanol showed a linear dependence with concentration (0–25 µM), showing that it is monomeric at concentrations of 100–200 µM, whereas in aqueous solution it was shown to aggregate at a concentration of 3 µM [64]. The composition of Nystatin includes three antibiotics, namely, Nystatin A1, A2, A3, and a small heptaene contaminant [65]. Nys consists of a 38-membered macrolactone ring with sets of two and four conjugated double bonds separated by a saturated bond (Figure 2). The molecule contains a mycosamine moiety linked to the macrolactone ring via a β-glycosidic bond, and exocyclic carboxyl group, both of which appear to be important for biological activity and toxicity [66].

AmB is the most important polyene macrolide antibiotic; it is a heptaene and contains a macrolactone ring that is β-glycosylated to a mycosamine group at the C19 position (Figure 3) [62]. This macrolactone ring is an almost planar chromophore with seven double bonds in trans conformation which make up the hydrophobic region; it also contains a hemiketal ring at the C13 and C17 positions. Additionally, it has a more flexible polyol subunit that is the hydrophilic section of the polyene. The amphoteric character of AmB is determined by the presence of a carboxyl group at the C16 position and an amino group located on the mycosamine head group [62].

At neutral pH, AmB is a zwitterion, a state resulting from the positively charged -NH3+ and negatively charged -COO-, which has an influence on the aggregation process [67]. The acidity of the solution can influence the molecular organization of AmB in an aqueous medium. At a concentration of 10 μM of AmB and a pH higher than 10 it remains in a monomeric form, whereas in a pH range 3–10 shows weakly coupled aggregated structures. Surprisingly, strongly coupled aggregated structures appeared at pH below 2 in spite of the positive net electric charge [68]. Aggregation is also concentration dependent, so the previous results could vary at other concentrations, or in the presence of other solutes. As mentioned above, AmB contains well-defined hydrophobic and hydrophilic regions, which gives it its amphipathic character. As a consequence, AmB is poorly soluble in water and tends to aggregate in aqueous solution [69,70]. It has been suggested that aggregation in solution might be important for the interaction with the membrane [71,72] and, recently, fundamental for pharmacological action [60,73,74,75]. From the pharmacological point of view, the aggregation state of AmB seems to produce highly toxic side effects in mammalian cells [72] given that such structures may directly assemble into porous structures in these membranes, affecting the physiological transmembrane ion transport [76]. This and other physicochemical properties of AmB have been critical in determining the proposed mode of action.

Another polyene that has been the focus of several studies is filipin 3. The term filipin refers to a compound that is a mixture of four isomeric forms (filipin I (4%), II (25%), III (53%) and IV (18%) [77]) of a polyene antimycotic synthesized by *Streptomyces filipinensis* [78]. The structure of filipin III was determined in 1995 by reporting its relative and absolute stereochemistry. Filipin has a structure that differs slightly from AmB and Nys due to a shorter polyene chain, consisting of five conjugated double bonds; it also lacks the mycosamine sugar moiety (Figure 4) [79]. Filipin has an antifungal activity, though it is not selective between fungal and mammalian cells [80,81,82,83].

The thermostability of filipin in the crystalline state was investigated by Tingstad and Garrett [84]. The half-life of filipin in air at 70 °C is about six hours; the molecule is 50 to 100 times more stable in the absence than in the presence of air and the loss of its biological activity correlates directly with these characteristics. Filipin is a membrane disruptor and is used to locate cholesterol in cell and lipid membranes [85]. However, the data must be interpreted with care [86] as cholesterol diffusion can lead to dissociation of the complex. It has also been used to detect Niemann–Pick type C disease [13] which is an autosomal recessive lysosomal storage disorder due to mutations in the NPC1 or NPC2 gene that can lead to a fatal disease in neonatal infants or chronic neurodegeneration in adults [87]. Filipin, similar to other polyenes, has been used to construct semi-synthetic derivatives searching to increase safety [88].

Natamycin 4 (also called pimaricin) is another effective polyene antibiotic with a large record of applications [89]. It is produced by *Streptomyces natalensis* and is used for the topical treatment of fungal infections. It is effective against a broad variety of yeast, some protozoa, and some algae [90] and it is also widely utilized in the food industry to prevent mold contamination of cheese and other non-sterile and fermented foods due to its selective action against yeast and mold and its inaction against bacteria [89,91,92]. Natamycin consists of a macrocyclic lactone ring with four conjugated carbon–carbon double bonds (tetraene) and a mycosamine group that renders it amphoteric (Figure 5) [90].

The amphoteric character of natamycin is responsible for its low solubility in most solvents [93], as is the case for AmB. Previous studies have reported that, despite being a selective polyene like Nys or AmB, Natamycin’s mechanism of action is not linked to membrane permeabilization despite its molecular similarities. This begs the question of whether other polyenes have a similar activity—one not related to the membrane pores, but simply sterol binding [92].

### 2.2. Oxidation

Polyenes are readily auto oxidized. For instance, AmB reduces its chromophore chain to a pentaene, as presented in Figure 6 following Gagos and Czernel [94].

Oxidation drastically modifies the absorption spectra and can lead to confusion with the aggregation state of the polyene since this phenomenon also affects the aforementioned absorption spectra. Aggregation is important for understanding distinct modes of action and must be determined carefully. Oxidation also has an important effect on pharmacological activity. For instance, AmB oxidation reduces its antifungal properties 16-fold against two *Candida* strains and affects its cytotoxic activity towards GMK cells 5-fold [95]. The pore mechanism of action assumes that the chromophore chain is essential for the formation of the barrel-like structure; an alteration of this chain would affect the formation of pores, and thus fungal and cytotoxic action. Oxidation occurs readily when the polyene is in solution. For example, in Figure 7, we presented the absorption spectra of AmB as time passes in PBS (ambient conditions in open vessels). As can be seen, oxidation in the dark appears to be noticeable after 24 h. Furthermore, light is promoting oxidation. The same figure shows that 96 h illumination with a LED lamp (spectrum from 400–700 nm with peaks at 450 and 550 nm) at 2000 luxes conduces higher oxidation vis-à-vis the same ambient conditions without light.

### 2.3. Aggregation

The activity of polyene antibiotics depends on their aggregation state [96]. There are two commonly used techniques for determining aggregation of polyenes: Circular Dichroism (CD) and UV/vis absorption spectra. In certain solvents (methanol, propanol, and dimethyl sulfoxide (DMSO)) polyenes exist as monomers, whereas the antibiotics are found to aggregate in aqueous solutions, particularly in PBS. The aggregation involves hydrophobic interactions [69]. Polyenes are mixed in deoxycholate to favor solvation for clinical application. However, some, like AmB, lose the zwitterionic character at low pH values of 3 and change into an ionic form which is quite soluble in water. In this case, the profile of the absorption spectra remains unchanged with concentration, similar to the case of solvation in methanol solution. Something similar occurs at a high pH value 10 but, at a physiological pH in PBS, the situation is quite different: the AmB absorption spectra for concentration below 0.2 µmol show that the polyenes are at the monomeric form [17,72]. At concentrations larger than 0.2 µmol, the profiles keep changing. Now, the peak at 347 nm keeps increasing, indicating aggregation; it starts from dimers and grows to larger aggregates [72,97]. Hence modification of the absorption spectra allows for the detection of the aggregation of polyenes [98,99,100], but one has to be careful because oxidation also produces changes in the same range of the spectra. To avoid oxidation, tocopherol is added to aqueous solutions [101].

Aggregation in solutions has been considered an important factor determining the absorption and insertion of polyenes into the membrane [71,72,102]. Even small aggregates, such as the dimer, are seen as crucial in the adsorption (or no adsorption) into membranes of different compositions. They are believed to be the reason for the different action in ergosterol- or cholesterol-containing membranes. Furthermore, the threshold at which dimerization occurs has been advanced as a reason for derivative improvement [72], which has prompted interest in studying the dimerization of AmB and its derivatives [72,103,104]. All simulations predict ready aggregation leading to dimerization and then a continuing aggregation, in agreement with the absorption spectra observations. The dimer geometry has been observed with different monomer orientations, but no case showed a hydrophobic area reduced upon aggregation, as could be expected. This suggests why aggregates could insert better into the membrane. Understanding what sort of molecular interaction leads to aggregation would help in derivative design. However, in this case, we have discrepancies. Are they due to dipole–dipole interactions or to hydrophobicity? The problem is that molecular dynamics simulations depend crucially on the force fields used, and the existing force fields for these molecules have not been as fully tested and validated as other molecules. For instance, the ΔG of solvation predicted by Zielińska et al. [104] is −7.1 kcal/mol, a value far too large for a molecule that is quite insoluble in water, less than 1 μM. Given the very useful information MD studies could yield, a refined molecular dynamics simulation is called for.

It has been suggested that very large aggregates of polyenes in aqueous solution exist and occur on the surface of membranes. Milhaud et al. [105] observed, by Atomic Force Microscopy, that very large aggregates of AmB were formed at the surface of a L-dilauroyl phosphatidylcholine membrane and that the morphology of these aggregates was dependent on the presence of ergosterol. Likewise, large aggregates of nystatin have been reported on membrane surfaces [106]. A mechanism of action based on the existence of large aggregates on the cell membrane was recently proposed [60,74,75]. It seems clear that large aggregates can occur; what is not clear is at what concentrations and, most importantly, at what polyene-lipid ratio they appear. It seems unlikely that they would occur at the 1 μM concentration—that is, the therapeutic active concentration.

Given the collateral toxicity of polyenes, new ways of bypassing this problem have been addressed, including the production of derivatives with the same effectiveness but less collateral damage.

## 3. Mechanisms of Action

Polyene antibiotics are thought to act mainly at the membrane level, though some have suggested they do so at the intracellular level, where the cell membrane is not directly involved in cell death (for instance through oxidative damage) [107,108,109,110,111]. It has also been suggested that oxidative bursts result from an AmB interaction with a membrane enzyme, NADPH [112,113]. Mousavi and Robson [114] treated protoplasts of *Aspergillus fumigatus* with hydrogen peroxide and AmB. In both cases, apoptotic-like phenotypes were detected, in contrast to pathogens treated with itraconazole, a fungistatic agent, which did not present these phenotypes. Blum et al. [115] found that *Aspergillus terreus* had a higher production of catalase, an antioxidant agent, than did the susceptible strand *Aspergillus fumigatus.* Hence, the authors hypothesized that the resistance could be due to a reduction in the oxidative damage caused by AmB. Sharma et al. [116] found that treating *Candida* isolates that presented reduced antifungal sensitivity with polyphenol curcumin I in combination with AmB reduced the minimum inhibitory concentration and that this combination was associated with the production of reactive oxygen species.

Another mode of action that has drawn attention is AmB’s potential effect as an immunomodulatory drug [46,117]. This process is not yet well understood, but the existing evidence points to AmB binding to Toll-like receptors (TLR) in the membrane. After binding, an adaptor protein, MyD88, is recruited. This eventually leads to macrophage activation. Moreover, the immunomodulatory effect depends on the formulation in which AmB is delivered. For example, in a study using AmB in deoxycholate, liposomal and colloidal dispersion formulations, and plasma of human patients, it was found that AmB in deoxycholate and liposomal formulations increased levels of proinflammatory cytokines while the colloidal dispersion formulation did not [118]. Another study using human monocytes showed that AmB in deoxycholate and in colloidal dispersion up-regulates inflammatory cytokines, while AmB in lipid complex and liposomal formulations down-regulates or has no effect on the expression of these proinflammatory cytokines [119]. This is a very interesting aspect of polyenes that has not been exploited at large because of the collateral toxicity they exhibit. That said, the development of polyene derivatives with increased safety that maintain immunomodulatory properties could have a strong impact on health.

The idea that polyene activity is related to membrane structure is also present in the super lattice model of sterol within the membrane. The model hypothesizes that sterols distribute themselves in hexagonal superlattices at specific sterol mole fractions that seem to be periodical and that Nys binding to lipid bilayers is correlated to the presence of these structures [120,121,122,123,124].

We now concentrate on more traditional models where action presumably occurs at the membrane level. In filipin’s case, a mechanism was proposed by Kruiff and Demel in 1974 [54]; their hypothesis is that filipin forms aggregates of approximately 15–25 nm in diameter within the membrane core. This aggregate is presumably composed of parallel arrays of filipin, stacked one above the other. Furthermore, these aggregates can form complexes with cholesterol that are totally embedded in and covered by the membrane; these complexes were visualized using electron microscopy, which showed that, on the edges of these aggregates, there is a strong curvature of the lipid layers of the membrane. The authors hypothesized that a fragmentation of the membrane takes place in here. Using atomic force microscopy (AFM), Santos et al. [125] also observed these protrusions in dipalmitoyl phosphatidylethanolamine (DPPE)/cholesterol-supported lipid bilayers of mean diameter of 19 nm and height of 0.4 nm as well as doughnut-like lesions and larger circular protrusions of mean diameter of 90 nm and 2.5 nm in height. These results showed that filipin action on DPPE bilayers is affected by cholesterol concentration. Lawrence et al. [126] performed AFM imaging of sphingomyelin/1,2-dioleoyl-sn-glycero-3-phosphocholine (SM/DOPC)-supported bilayers, either sterol-free or with 10 mol% cholesterol, and observed filament-like aggregates of filipin on high-order SM-rich domains only in the cholesterol-containing bilayer. This suggests that SM-domains are also enriched in cholesterol. The filament-like aggregates showed a periodicity of ~4.3 nm. The lack of lesions or aggregates on the sterol-free bilayer suggests that not only lipid physical state (e.g., gel) but also lipid species comes into play in the interactions between filipin and lipid bilayers. In Santos et al. [125] and Castanho and Prieto [127], for example, there was no need for sterol. We must consider that the above studies with supported lipid bilayers entailed high concentrations of filipin (~100 uM); the polyene lipid ratio was also quite high.

The oldest and most accepted mode of action is that polyenes form membrane spanning pores that cause an electrolyte imbalance that leads to cell death [54,56,128,129]. In 1968 [18], both Nys and AmB were shown to radically reduce the lipidic membrane resistance to direct current and thus alter the ion selectivity properties of cholesterol-containing lipid membranes. This led to the hypothesis that the interaction of these polyene antibiotics with cholesterol bound to the membrane modifies the surface properties of said membranes and, consequently, their electrical behavior. Nys and AmB also induced permeability to water and non-electrolytes, so it was suggested that both polyenes create aqueous pores in lipid membranes with similar permeability [130]. These pores do not appear to be static or permanent, they have a strong temperature dependence. Furthermore, the increase in lipid membrane conductance due to these pores was directly related to the increase in polyene concentration [11]. The pores were thought to be barrel-like structures composed of polyene and sterol molecules [54,56,128,129]. A pore model based on the chemical structures of polyenes and cholesterol was proposed, and consisted of a circular arrangement of 4 to 12 polyene molecules interdigitated with cholesterol. The interior was hydrophilic due to the presence of hydroxyl groups, and the exterior was hydrophobic [54,56,129], where the apolar backbones of the polyene and sterol were oriented parallel to the fatty acid chains of the phospholipid. The total length of this complex was approximately equal to the length of the fatty acid and glycerol moiety of the phospholipid molecules in such a way that a single complex was a half-pore through the lipid bilayer. Two such half-pores were thought to be needed on each side of the lipid bilayer to obtain a complete conducting pore [54,131]. It was recently revealed using solid state nuclear magnetic resonance (NMR) that the AmB-Erg complex is able to span the whole bilayer via thinning of the membrane. The authors complemented these experiments with a molecular dynamics simulation to show the thinning of the bilayer [132]. It can be assumed that transmembrane pores are created in two different ways: either by a polyene-induced conformational thinning effect in the lipid bilayer, or by the union of two half pores without the need for changes in membrane conformation. It has been shown that these two structures (single and double pore) exist and present different ion selectivity [133]. Kruijff and Demel [54] determined that the diameter of AmB’s pore is about 8 Å by observing the largest molecules that permeated the membrane (glucose). In the case of Nys, the pore is slightly smaller in size, which could be explained by the bending of the hydrophobic backbone of the nystatin molecule. The size of the pore radius depends mainly on the structure of the polyene, the number of monomers (between 4 and 12 molecules), and the participation of other membrane molecules, such as sterols [17,63,134,135]. All the previous studies were indirect, and it was not until the advent of single-channel electrophysiology experiments of polyene action on lipid membranes that the existence of the pore was confirmed. Early single-channel electrophysiology studies of AmB and Nys studied conductance changes in planar lipid bilayers related to single pore formation [136], where stepwise changes in conductance were observed as a result of the formation of single channels, each with two states, one open and one closed. Single-channel electrophysiological experiments also allow for the description of channel kinetics. After its formation, the channel exhibits many transitions between open and closed states. The average dwell time in each state does not depend on the membrane potential, but on the concentration and type of salt present. The channel kinetics of Nys and AmB were shown to be similar, with the only difference being smaller jumps and somewhat longer dwell times for open channels in the case of Nys. Furthermore, more recent studies have shown that the pore is a supramolecular structure [137] with differing channel size depending on polyene concentration and, thus, the number of monomers present in the structure.

As previously explained, sterols play an important role in the pore formation model. Sterols are thought to be responsible for pore stability, with ergosterol providing more and thus allowing for larger pore size [138] and longer open dwell times [137]. The pore-forming model has been modified and enriched over the years. For instance, it has been used following the alternative idea that membrane selectivity (i.e., the polyene being more effective in fungal membranes containing ergosterol than in mammal membranes containing cholesterol) is due to the membrane structure differences produced by sterols, rather than a polyene–sterol direct interaction.

The fact that AmB channels can be formed in lipid bilayers that do not contain sterol provides experimental evidence supporting the idea that sterol is not an absolute requirement for pore formation [135]. For this to occur, however, larger concentrations of AmB were needed. It has been suggested that sterol-free structures correspond to non-aqueous pores that will not evolve into aqueous ones, and are therefore not pharmacologically relevant [139]. However, it has been shown that the molecular properties of the sterol-free single channels suggest supramolecular structures similar to those found in the presence of sterol [135]. Evidence supporting the idea that membrane structure differences are the reason behind selectivity was presented in a study of Nys channel activity [58] along a previously reported phase diagram for POPC/ergosterol and POPC/cholesterol lipid mixtures [140,141]. Nys was found to present higher activity in conditions where there is phase coexistence—that is, liquid ordered and liquid disordered phases. Recently, further evidence that polyene activity can occur in membranes without sterols but in particularly ordered conditions was presented [59].

There are still many ongoing studies looking into the properties of these pores by electrophysiological techniques as well as other approaches. For instance, there are studies of the properties of POPC giant unilamellar vesicles (GUVs) under osmotic stress produced by Nys in the presence of cholesterol, ergosterol, or sterol free [142]. This work makes clear that, even in large Nys concentrations, the GUVs with ergosterol do not resemble the properties of the sterol-free GUVs, as would occur if the polyene was extracting the sterol. On the other hand, ergosterol-containing GUVs present an osmotic effect that indicates a larger amount of Nys pores. In a recent study combining NMR and MD [132], it was suggested that only the classic pore model of AmB and ergosterol could explain the observations in a POPC membrane. Another recent work [143] using polarization-sensitive stimulated Raman scattering found that AmB resides inside the cell membrane and is highly ordered, with an orientation primarily parallel to phospholipid acyl chains, supporting the channel model. Recent electrophysiological studies of channels in lipid bilayers have been performed for different compositions and conditions and have shown a variety of behaviors. For instance, bilayers of 1,2-diphytanoyl-sn-glycero-3-phosphocholine with a large content of cholesterol have presented large conductance channels of AmB [61]. These conductances can reach very high values (~400 pS) when the bilayer is formed on a nanoporous solid support made of silicone dioxide, highlighting the strong dependence of channel architecture on environmental conditions. Similarly, the addition of dipole modifiers to a bilayer presenting AmB channels [144] considerably modifies the single channel conductance and opening times of AmB channels.

Polyene activity in the absence of sterol was also found to be produced by osmotic pressure [145]. These authors found that AmB activity, resulting in increased membrane permeability, is observed in sterol-free LUVs. They found a relation between activity and osmotic pressure changes that produce higher curvature. It is worth mentioning, however, that the study was not able to determine whether such activity was linked to channel formation or to membrane disruption. Still, it is stated that polyenes have a differentiated action due to structural differences in the lipid bilayer. In this study Wolf and Hartsel [145] advanced the idea that polyene membrane penetration is the critical factor determining membrane selectivity. They suggested that polyenes may act as what is called Molecular Harpoon (MH) [146]. This concept is used to describe certain amphipathic compounds that are easily inserted into lipid membranes. These molecules, once inserted, induce instability of the membrane, leading to the permeation of ions, but not necessarily leading to membrane rupture. The insertion of MHs is strongly related to the state of oligomerization of these compounds [145]. Permeation induced by MHs is enhanced when the bilayer is subject to osmotic stress. In this case, lower concentrations of MH were required [146,147]. The hypothesis was that the greater ease of insertion is due to the formation of crevasses in the external monolayer, exposing the hydrophobic core [146,147,148,149].

We know that the activity of MHs can be affected by the physicochemical properties of the membrane. For instance, the activity of triton X-100 and other synthesized wedge-shaped surfactants is affected by cholesterol content and osmotic stress [147]. Both are presumed to affect the lipid packing of the membrane, which in turn favors or not the insertion of MHs. Wolf and Hartsel [145] observe that AmB and Nys at 5 μM do not present appreciable ion permeation in LUVs without sterols, but the activity takes place when these LUVs are subjected to osmotic pressure. This activity is dependent on osmotic pressure magnitude. This model is related to one of the processes involved in the polyene mechanism of action—insertion into the membrane—and thus would be worth revisiting. For instance, many of the previously mentioned phenomena could be explained by a difference in membrane insertion that is affected either by composition or physicochemical properties of the membrane.

To finally elucidate the mechanism by which AmB and other polyenes form pores or other structures in the membrane would require observation at a molecular level. In this regard, molecular dynamics simulations could prove a very important tool and the literature includes studies using this technique [138,150,151,152,153,154,155,156]. For example, molecular dynamics simulations of the pore suggest that ergosterol better stabilizes it, allowing for larger pore dimensions [138]. This stabilization is thought to be due to a direct interaction between ergosterol and AmB, which was observed via ^2^H NMR spectra of deuterated sterols in a palmitoyl-oleoyl-phosphatidylcholine bilayer [157]. However, there is no standardized potential for AmB and other polyenes [104,153], which posits a problem and renders the results somewhat unreliable. This becomes more problematic because ergosterol itself has a force field potential that is still under constant refinement [158]. Furthermore, in the studies that look into the dynamics of AmB structures on the membrane surface, the initial conditions presume particular constructions that would not dissociate because of large potential barriers that have been overcome by hand. The ideal simulation would need validated potentials for both polyenes and lipids, and the stepwise addition of polyene monomers to the aqueous phase of the membrane system. This latter case was considered in a recent article [159] where the pore was not constructed as an initial condition of the simulation. Polyene monomers became aggregated in the membrane, showing for the first time the spontaneous formation of the pore. However, this study was performed using coarse-grained molecular dynamics and we therefore lack a total atomic description of the system.

Finally, a model of action that has gained much recent popularity is the so-called sponge model [60,73,74,75,160], according to which AmB molecules are adsorbed on the membrane surface in the form of large aggregates, where the polyene molecules are in a perpendicular orientation to the normal of the lipid bilayer. This aggregate acts as a sponge that draws ergosterol from the lipid membrane, keeping it inside the polyene aggregate due to the high binding affinity between AmB and ergosterol, causing interference in crucial functions for fungal cells [60]. However, this model is still under intense scrutiny. A recent study using polarization-sensitive stimulated Raman scattering from the C=C stretching vibration of the fingerprint to image AmB found that, in 16 different fungal cell types, the orientation of AmB paralleled the orientation of the lipids [143]. Another study using solid-state nuclear magnetic resonance found further evidence for parallel orientation of an AmB derivative within the lipid membrane [161]. Both results support the barrel ion channel model. Kaminski [44] pointed out that the ergosterol–lipid ratios used in two of the main works supporting the sponge model [60,75] are significantly lower than those present in real cell systems like *Saccharomyces cerevisiae*, namely 1:10 and 1:40 in the studies in comparison to 3:7 in *S. cerevisiae* [162]. Kaminski also pointed out that in an environment where cholesterol is available in larger amounts than ergosterol, as in a human host infected with a fungal pathogen, the balance between AmB–ergosterol and AmB–cholesterol interactions will favor the latter, meaning that ergosterol extraction from the fungal cells will be greatly diminished. Finally, Kaminski made an argument that the fungal cell wall is made up of hydrophilic chitin, which poses a serious obstacle in transporting ergosterol from the cell membrane and into the polyene aggregates. Additionally, there is evidence that not only is the affinity lower for cholesterol, but that AmB does not extract it from POPC membranes [163]. The authors used neutron reflectometry and found that AmB is not capable of extracting cholesterol from a POPC-cholesterol lipid bilayer, but it does extract ergosterol from a POPC-ergosterol lipid bilayer.

Polyene selectivity towards sterol type is at the basis of their therapeutic use. This selectivity, however, is not as marked as to prevent large collateral toxicity, therefore limiting the extended application of polyenes in a variety of pathologies. This has led to intensive research on how to make polyene use safer, either via derivatives or new formulations.

## 4. Alternatives to Reduce Polyene Host-Toxicity

### 4.1. Polyene Semi-Synthetic Derivatives

AmB, along with other polyene antibiotics, has an excellent antimycotic effect as well as broad antifungal spectrum; unfortunately, it is highly toxic, which often limits its effective use as a last line of defense against life-threatening systemic fungal infections [164]. AmB in particular is considered the most effective drug for the treatment of systemic fungal infections [24,165]. For this reason, the semi-synthesis of new derivatives has been focused on this drug [166,167,168,169]. It is not easy to chemically modify AmB given its dense array of functional groups, e.g., the macrolactone is susceptible to saponification, the heptaene and the hydroxyl groups are prone to oxidation, and the hemiketal and the mycosamine are acid-sensitive [170]. The C16 carboxy group is an easily accessible locus for chemical modification by esterification or amidation.

Now we present successful derivatives. That is, derivatives that keep the antimycotic efficiency of AmB and considerably reduce its host toxicity.

Paquet and Carreira [171] documented the synthesis with significant improvement in antifungal activity via double reductive alkylation of the mycosamine—that is, the introduction of two amino propylene sidechains to produce the *N*,*N*-di-(3-aminopropyl) AmB derivative **5** (Figure 8). This compound exhibited significant inhibitory activity against an AmB-resistant *Candida albicans* strain with a MIC value of 4.0 µM, and was also more active against *S. cerevisiae* than AmB with MIC of 0.10 µM. In hemolysis assays, **5** displayed less toxicity for blood cells, EHB_50_ 50 µM, compared with AmB (EHB_50_ 4.0 µM).

Another successful compound was produced by modification of the C16 carboxy group of AmB. The amide *N*,*N*-dialkyl derivative **6** (Figure 9) that was synthesized and evaluated by Preobrazhenskaya et al. [172] exhibited a high activity against four fungal strains and lower hemolytic activity compared to AmB for in vivo studies. The minimum inhibitory concentration (MIC_50_) was 0.08 µg/mL compared with AmB’s 0.11 µg/mL. On the other hand, it showed acute toxicity in mice, with a lethal dose (LD_50_) of 16.4 mg/kg compared with AmB’s 2.8 mg/kg.

Davis et al. [173] synthesized AmB urea derivatives in just three steps from AmB. In vitro tests showed a very successful compound **7** (Figure 10) with considerable increased safety. The authors proposed that the increased membrane selectivity was due to its binding to ergosterol while doing so less effectively to cholesterol than AmB. This was shown using isothermal titration calorimetry (ITC).

Another successful product is *L*-histidine methyl ester, derived from Amphotericin B (**A21**) **8** (Figure 11), which has an *L*-Histidine that substitutes the carboxyl group of AmB and was shown to be less toxic than AmB in in vitro and in vivo tests [72]. The authors hypothesized that its greater safety is due to a smaller dipole moment that reduces the aggregation threshold.

In 2020, Tevyashova et al. [174] synthesized a series of AmB derivatives with presumably reduced aggregation properties and designed a series of C16-carboxamides of AmB containing a basic group that can be protonated and cause reduced aggregation in aqueous solutions as well as improved water solubility. Previously this same research group studied a series of semi-synthetic genetically engineered derivatives, proving that the introduction of a side chain with a tertiary amino group on the amide moiety led to improved water solubility and, in some cases, to an increase in the antifungal activity of derivatives [172,175,176]. The introduction of the positively charged group at the C16 position also disrupts the zwitterionic interaction between the carboxy group of C16 and the amino group of mycosamine, increasing the solubility of the compounds [174]. They also reported a series of derivatives obtained by the transformation of C16-carboxylic group into carboxamide. The molecule obtained from 1,2-diaminoethane and AmB in particular demonstrated a higher antifungal potency than that of parent AmB. The *N*-(2-aminoethyl) amide of AmB **9**, which they called “amphamide” (Figure 12), has an ionic form that is more stable and soluble in water. It has a considerably increased safety and efficacy compared with those of AmB, with a therapeutic index calculated as the ratio between the lethal dose and the effective dose (LD_50_/ED_50_) of 41.8 in a murine model.

Unfortunately, despite all of the above-mentioned efforts and success in synthesizing a polyene derivative with equal efficacy and less toxic effects, none of these AmB derivatives are yet in clinical use. Another alternative to reduce polyene host toxicity is to develop new formulations that deliver the drug in a more precise manner. Several formulations of AmB are now available for therapeutic use [177] and, in the case of Nys, a lipid formulation is in phase III clinical trials [178].

### 4.2. Lipid-Based Formulations

In the past decades, much effort has been made to develop and use new AmB formulations with equal efficacy but lower host toxicity. These include lipid-based formulations to deliver polyenes, such as liposomes or lipid complexes [24,117]. In addition, other forms of delivery consider emulsions [179,180], polymeric nanoparticles [181], and a nanoparticle-based encochleated AmB oral formulation [182]. Here we focus on lipid complexes and liposomal formulations.

Since the 1990s, some of these formulations went into clinical use: AmB lipid complex with the commercial name Abelcet^®^ [183,184], which is formed in a ribbon-like shape [185]; there is also a colloidal dispersion of AmB and cholesteryl sulfate that forms disk complexes and is sold under the commercial name of Amphotec^®^ or Amphocil^®^ [186,187,188,189,190]; a liposomal formulation of AmB in cholesterol-containing lipid vesicles sold under the name of Ambisome^®^ [191,192,193,194]; and a multilamellar vesicle formulation, with a different lipid mixture and sold under the name Fungisome^TM^ [195,196,197,198,199,200]. In the case of Nys, a liposomal formulation called Nyotran^®^ [201,202,203,204,205] is currently undergoing phase III clinical trials [178], and the available results are promising. Lipid formulations have distinct pharmacokinetic profiles and thus have specific dose and administration requirements [206,207,208,209,210,211,212,213,214]. They are supposed to target specific cell membrane properties or the cell wall of the fungal pathogens. The latter case was observed by Walker et al. [215] by means of electron microscopy. The authors found that the viscoelastic properties of the fungal cell wall seem to favor the traffic to the plasma membrane of AmB-loaded vesicles, whereas unloaded vesicles do not cross the cell wall. The authors hypothesize that AmB-loaded vesicles cross the cell wall due to AmB’s binding to ergosterol in the mannose filaments due to exosome transit through the cell wall. We believe that, as the authors themselves showed, all liposomes reach the outer cell wall, where we think the AmB-loaded liposomes release AmB into the medium, contrary to the authors’ idea that delivery only occurs when the liposome fuses to the plasma membrane. In our idea, AmB reaches the plasma membrane and destabilizes it along with the fungal cell wall, changing its porosity and allowing for the eventual transit of the liposomes towards the plasma membrane.

Lipid formulations can also penetrate fungal biofilms in a more efficient way than the usual AmB-deoxycholate formulation [216,217,218,219,220]. It should be noted that most lipid formulations for polyenes make use of high transition temperature lipids (e.g., hydro soy PC in AmBisome^®^ or DMPC in Nyotran^®^). This is important for two reasons. One is that high transition temperature lipids seem to form stable polyene:lipid complexes in comparison to low transition temperature lipids [59,64,221,222], which, along with the presence of cholesterol, favors the incorporation of AmB into the lipid membrane. The second reason is that high transition temperature lipids confer the liposomes with enough physical stability to prolong their blood circulation half-life [223]. Another important factor that contributes to the efficacy of AmBisome^®^ is the small size of the liposomes, which allows for the evasion of the reticuloendothelial system that clears blood plasma from large particles. This fact has been further reinforced by the results obtained by generic versions of AmBisome^®^ against the original formulation, where the former have a larger particle size and show a lower efficacy in the treatment of fungal infections [117]. Lamellarity is another critical parameter, as Fungisome^TM^, which is stored as multilamellar vesicles for better stability, must go through an ultrasonication step prior to infusion in order to obtain unilamellar vesicles from the multilamellar ones [200]. Electric charge in liposomes, e.g., in the form of phosphatidylglycerol (PG), also plays a role in the efficacy of lipid formulations. Liposome–cell interactions depend on the surface charge present in the liposome bilayer, which can be neutral, positive, or negative [224,225]. Furthermore, neutral liposomes will tend to aggregate and thus have a reduced physical stability [226], along with a tendency to release their cargo away from the target cell given both instability and a lower liposome–cell interaction [227]. Finally, the process of liposome manufacturing follows critical parameters, e.g., acidification, liposome heat curing, etc., that must be carefully carried out to obtain the best quality final product [228].

New lipid formulations are currently under research and development and exploit new and exciting properties such as surface modified liposomes [229] and even liposomes with encapsulated iron oxide yielding magnetic properties [230]. For a more detailed review on this topic, we recommend Faustino and Pinheiro [117]. As we have seen, the action of polyenes is closely related to the membrane properties, whether in solution or liposomal preparations. Hence, an understanding of membrane properties and its involvement in polyene action are crucial.

## 5. Role of Membrane Structure on the Activity of Polyenes

### 5.1. Binary Lipid Mixtures Containing Sterol

Binary mixtures consisting of a lipid species and a sterol are the simplest study model for the effects of sterol on membrane properties. While ergosterol has been less studied than cholesterol, we can obtain information regarding their similarities and differences from the available literature. The addition of ergosterol to DPPC-ordered bilayers produced a phase separation of gel- and liquid-ordered phase, as well as filament-like structures on both of them. This was observed via AFM of supported lipid bilayers (SLB) [231]. Cholesterol presence on DPPC bilayers produces much larger domain-like gel-phase regions that contain smaller liquid-ordered regions, as observed on Langmuir–Blodgett monolayers imaged using AFM [232]. Thus, both sterols produce segregation of gel and liquid-ordered phases on DPPC bilayers, though their morphologies vary. Furthermore, polyenes affect membrane structure. Using a Langmuir trough to obtain lipid monolayers and Brewster angle microscopy, as well as atomic force microscopy, Wang et al. [233] showed that AmB has an effect on the packing of lipid and polyene molecules in DPPC bilayers, either sterol-free or containing 30 mol% cholesterol or ergosterol. The authors also found, by using the limiting molecular area analysis, that depending on the sterol used, AmB affects the monolayer containing an unsaturated lipid, DOPC, differently. This difference is smaller for monolayers containing saturated lipids, DPPC. Finally, the authors hypothesized that AmB could orient itself differently when it inserts in lipid/sterol bilayers depending on the saturation of the lipid involved, which could help understand the toxicity towards cells. A Deuterium Nuclear Magnetic Resonance (^2^H NMR) and Differential Scanning Calorimetry study showed that ergosterol induces less liquid ordered domains than cholesterol, in both gel- and liquid-disordered DPPC membranes [234]. In 1-palmitoyl (2H31)-2-oleoyl-sn-glycero-3-phosphocholine (POPC, d_31_) multibilayer vesicles, lipid ordering was evaluated using ^2^H NMR spectroscopy for different sterols, including cholesterol and ergosterol [235]. The study found that all sterols increase lipid chain ordering at increasing concentrations but have distinct limits in this ordering, with cholesterol having a higher limit than ergosterol. The authors hypothesize that the C22 double bond on ergosterol could be the underlying cause for its lower limit. In a study using X-ray Scattering and Grazing-Angle Scattering, lipid bilayers of a saturated lipid, 1,2-dimyristoyl-sn-glycero-3-phosphocholine (DMPC); a mixed alkyl lipid, (POPC); and an unsaturated lipid, DOPC with either ergosterol or cholesterol showed differences in their condensing effect [236]. Ergosterol did not present a condensing effect on POPC and DOPC, but did so on DMPC, albeit to a smaller degree than cholesterol. This suggests that ergosterol does not have a thickening effect on POPC or DOPC. The authors suggest this could be due to a difference in the sterol-lipid interactions, where cholesterol has a hydrophobic-matching effect and ergosterol does not. However, Pencer et al. [237] used small-angle neutron scattering measurements on DMPC vesicles with either cholesterol, ergosterol, or lanosterol and showed that all three sterols increase bilayer thickness in a similar way. The area expansion coefficients were different for each sterol, indicating a difference in the condensing effect, with cholesterol having a higher effect than ergosterol. Although there are mixed results, there seems to be a difference in the effect of ergosterol and cholesterol on simple binary lipid mixtures. This explains the previously presented results on the action of Nys on POPC/sterol [58,238] lipid mixtures along a phase diagram [140,141].

Another structural model suggests that sterols distribute themselves regularly at certain periodical mole fractions. This comes from a study using fluorescence microscopy on dehydroergosterol-containing DMPC multilamellar vesicles [122] and cholesterol-containing DMPC and SM multilamellar vesicles [123]. The authors link this regular distribution to a hexagonal superlattice [121]. With this superlattice model in mind, they measured Nys’ partition coefficient in DMPC/ergosterol, DMPC/cholesterol, POPC/ergosterol, and POPC/POPE/ergosterol multilamellar vesicles at varying sterol content with small increments [120]. The authors found a correlation between Nys’ partition coefficient and proposed superlattice existence, suggesting a facilitated insertion into lipid bilayers due to the superlattice.

Clearly, sterols have a lot of influence on the action of polyenes in membranes. However, it is important to point out that even in the absence of sterols, polyenes modify membrane permeability to ions.

### 5.2. Sterol-Free Bilayers

Over the years, several authors have suggested sterol might not be absolutely necessary for polyene activity. One early study was that of HsuChen and Feingold [239], where AmB and Nys were shown to have different effects on the glucose release of liposomes made of egg, dipalmitoyl, or distearoyl lecithins with increasing mol% of cholesterol. The authors showed that, at 0 mol% cholesterol, there was glucose release for both polyenes. It should be noted that, in this study, the polyene concentration causing glucose release in the absence of cholesterol was low (5–10 μM), whereas more recent experiments required much higher polyene concentrations (100 μM) [135,137]. These latter studies found AmB single channels in lipid bilayers lacking sterol, including lipid extract from *Escherichia coli*. This discrepancy can be understood if we consider the polyene/lipid ratio for each study. In the first case the ratio was ~0.1, whereas in the second case it was ~0.01. This highlights the importance of considering said ratio when comparing results.

As mentioned previously, Harstel’s group performed several studies based on the idea that surface tension modified by osmotic pressure could have an effect on the activity of polyenes even in the absence of sterol [145]. Following these results, Ruckwardt et al. [240] performed similar experiments on sterol-free LUVs made of POPC, diecosenyl phosphatidylcholine (DEPC), and egg phosphatidylcholine. Under osmotic stress, LUVs made of POPC became more sensitive to AmB, even more than egg phosphatidylcholine. LUVs made from DEPC were unresponsive to AmB in spite of the applied osmotic pressure, showing that this effect is dependent on bilayer thickness and membrane composition. Furthermore, the concentrations of AmB used were within the therapeutic range (0.5–10 μM), and the polyene/lipid ratio between 10^−3^ and 10^−2^. Additional to osmotic pressure, lipid order seems to facilitate polyene incorporation and form AmB-lipid complexes. In monolayers of dipalmitoyl phosphatidylserine (DPPS) or DPPC, complexes of 2:1 (polyene: lipid) stoichiometry are thought to be formed [241,242]. Another study using monolayers [243] showed that in the presence of K^+^ AmB has a higher affinity for DPPC bilayers as compared with when Na^+^ is present.

Another possible mechanism of polyene action in the absence of sterol is phase segregation. Dos Santos et al. [59] used model membranes composed of a lipid with a high gel/liquid transition temperature (T_m_), either 1,2-dipalmitoyl-sn-glycero-3-phosphocholine (DPPC) or egg sphingomyelin (ESM), and POPC, a low T_m_ lipid with mixed acyl chains. At room temperature, both DPPC and ESM are in gel phase and form concentration-dependent, highly ordered domains. The authors argued that these ordered states facilitate polyene action in a sterol-independent manner. Their findings showed that the action of Nys is favored by the presence of gel phase domains given the moderate presence of membrane permeabilization in fluid membranes; this permeabilization rises with the increasing number of gel-phase domains. The results vary slightly between DPPC- and ESM-containing liposomes suggesting a lipid species dependence. Finally, the authors hypothesized that Nys’ interaction with the membrane seems to occur in the gel–liquid boundaries. These findings agree with previous results that suggest that AmB has a high affinity for the aliphatic chains of DPPC in the gel phase [244,245]. This agrees with the fact that filipin incorporates more in DPPC bilayers when it is in its gel phase [246]. Although this behavior might be polyene-dependent, as we mentioned previously filipin requires cholesterol to aggregate on SM/DOPC-supported lipid bilayers [126]. These results emphasize the fact that membrane structure is a determinant factor for polyene action on lipid membranes.

### 5.3. Ternary Lipid Mixtures Containing Sterol

Ternary lipid mixtures are a very interesting model system when they present different phase coexistence [141,247,248,249,250,251,252,253,254,255]. This allows for model membranes to better resemble natural cell membranes. Of particular interest are mixtures that present phase segregation and contain sterol. However, these models have seldom been used to study polyene interactions. Lawrence et al. [126] used filipin to detect cholesterol in SM/DOPC/cholesterol supported lipid bilayers imaged with AFM. They found that, in the sterol-free membrane, filipin had no effect on the lipid bilayer. However, the ternary cholesterol-containing bilayer filipin formed filament-like aggregates with a well-defined periodicity of ~4.3 nm. These filaments were only present on high-ordered SM-rich domains. This suggests that filipin binding to lipid bilayers is favored by gel-phase lipids and cholesterol presence. A recent paper [256] showed that adding ergosterol or cholesterol to a 1:1 mol/mol ESM/POPC bilayer had an effect on the morphology of the gel-phase domains. Increasing the amount of sterol reduced the coverage of gel-phase domains. This effect is more pronounced in cholesterol-containing membranes. At 20 mol%, cholesterol inhibits the formation of large gel-phase domains. Here we present additional studies of polyene binding to the previous ternary mixtures, with 20 mol% sterol where ergosterol-containing bilayers show significant domain coverage, whereas cholesterol-containing ones do not. Figure 13 and Figure 14 show the effect of adding either AmB or A21 to ergosterol- or cholesterol-containing ESM/POPC bilayers. Firstly, the required concentration of polyene to cause different effects in the topography of the bilayers is different for AmB or A21. The effect also depends on the particular sterol present. AmB has an effect on the ergosterol-containing bilayer at 1 μM, while A21 does so at 2 μM. In the cholesterol-containing bilayer, AmB shows some effect at 2 μM while A21 does so at 5 μM. Secondly, ESM domains swell for both polyenes acting on the ergosterol-containing bilayer, suggesting the insertion of both polyenes into the gel-phase domains. In the cholesterol-containing bilayer, where micron-size domains are non-existent, AmB causes bilayer damage in the form of nano-defects in addition to large membrane-free mica regions (indicated by the white arrows in Figure 14B). On the other hand, A21 seems to form aggregates on the bilayer surface and does not produce nano-defects. These results suggest that A21 and AmB act differently in cholesterol-containing bilayers, where phase segregation is almost non-existent, and somewhat similarly in ergosterol-containing ones, where there is clear phase segregation and gel-phase domains are large. This could help to explain the differences in the lower host toxicity produced by A21 on mice in comparison with AmB [72].

Another interesting effect is how polyenes change phase segregation on lipid bilayers of ternary mixtures. Chulkov et al. [257] used confocal fluorescence microscopy to evaluate how AmB, Nys, and filipin influence phase separation in giant unilamellar vesicles made of: the DOPC/SM/cholesterol (57/10/33 mol%) ternary mixture; the DOPC/cholesterol (67/33 mol%) binary mixture; and pure POPC or DOPC. None of these lipid mixtures produce phase segregation. In the presence of Nys, however, stable solid ordered domains are formed in the ternary DOPC/SM/cholesterol mixture, as well as in giant unilamellar vesicles made from the binary DOPC/cholesterol and pure POPC. The authors argued that each polyene’s ability to induce gel-phase domains correlates with each polyene’s biological activity, that is, their ability to increase membrane permeability.

The hypothesis that membrane structure is what drives polyene membrane selectivity has gained some momentum due to differences in raft characteristics between mammalian and fungal cells. Small, transient, ordered gel-phase cholesterol-enriched lipid rafts are thought to exist in mammalian cells [258]. In yeast cells, on the other hand, there are large gel-phase domains that might not be ergosterol-enriched [259]. This idea, as well as all previously discussed theories, could lead to new polyene derivatives and/or formulations with higher therapeutic index for clinical application. Clinical application is probably the greatest motivation for polyene study, and we shall therefore address some recent advances in this area.

## 6. Clinical Use of Polyenes

Clinical use of polyenes has been discussed thoroughly in previous papers [50,260,261,262]. Here, we present some United States of America Federal Drug Administration (FDA)-approved uses for polyenes that appear in the World Health Organization model list of essential medicines: 22nd list 2021 [21]. We then address their current clinical use and some important clinical advances.

In its several formulations, AmB is used to treat coccidioidomycosis via intrathecal route [263], American mucocutaneous leishmaniasis, invasive aspergillosis [264], blastomycosis [265], candidiasis [266,267,268,269,270,271], coccidioidomycosis [263,264,272], cryptococcal meningitis in patients with HIV infection [273,274,275,276,277], cryptococcosis [278,279,280,281], severe fungal infection of central nervous system [282,283,284,285,286], severe fungal infection of lung [287,288,289,290,291,292], histoplasmosis [293,294,295], histoplasmosis in patients with HIV infection [273], pulmonary cryptococcosis in patients with HIV infection [273,278], infection by Basidiobolus [296], mucormycosis [297,298,299,300,301,302], sporotrichosis [303,304], and severe urinary tract mycosis [267,305,306,307,308,309,310]. Nys is used to treat candidal vulvovaginitis [311,312], candidiasis of skin [313,314], cutaneous and mucocutaneous infections [315], and non-esophageal gastrointestinal candidiasis [316]. Natamycin is used to treat blepharitis (Fungal infection of eye), fungal conjunctivitis, and fungal keratitis [317]. In addition to polyenes’ use as antimycotics and antiparasitics, experimental trials of other therapeutic applications have been reported [318,319].

### 6.1. Recent Advances in Clinical Use

In recent years, the increased emergence of new diseases has wreaked havoc on the medical community and health systems worldwide, the most obvious example being the recent pandemic caused by the SARS-CoV-2 coronavirus (COVID-19) and its co-morbidity with pre-existing chronic diseases [320]. Problematically, public health policies in health institutes, research centers, and the scientific community have focused on the study of this disease, leaving aside other important conditions with high mortality and for which there are still no effective and safe therapeutic options, as in the case of diseases caused by fungi [321,322].

Fungal infections, particularly invasive mycoses, represent a serious problem for patients with compromised immune systems [323]. Invasive fungal diseases are still a major global health problem [324,325]. Because these diseases do not present a clear clinical picture, early diagnosis is consequently difficult and proper assessment only takes place once the disease is already very advanced and therapy no longer as effective. Recent studies show that the global morbidity and mortality of invasive fungal infections have substantially increased during the past decade [326]. About 1 billion people in the world are thought to suffer from a fungal infection, an increase due to host factors as well as new mechanisms of virulence or resistance. Additionally, recent environmental and epidemiological evidence of endemic mycoses shows changes in the geographic prevalence of pathogenic fungi worldwide [327,328]. It has recently been suggested that invasive mycoses are undergoing etiopathogenic changes that are making these diseases even more prevalent, and that the geographical distribution is moving towards geographical areas where it did not exist before [329]. The former matter is at least partially due to new risk factors such as the rise of new medications that alter the immune response and the appearance of new strains of pathogenic fungi, such as *Candida auris*; the latter is tied to phenomena such as human migration, new practices in agriculture, occupational exposure, soil movement, and climate change, which are significant triggers in the spread of the disease [330]. The treatment of invasive fungal diseases continues to lag behind despite available drug therapies. This is because of three key issues: (1) treatment costs are not accessible to all sectors of the population, (2) available medications might be too toxic and their use should be discontinued, and (3) health care systems are not well-equipped to handle silent diseases such as invasive mycoses given our aforementioned lack of tools for early diagnosis and the high treatment costs once diagnosis has been made. All of these factors play a role in the increase in morbidity and mortality.

Invasive fungal diseases are quite costly. They are serious illnesses that tend to become chronic, and this takes a toll on the economically active population. They entail substantially expensive lab tests, long hospital stays, treatments that must be administered for long periods of time, and lastly, very expensive medication [331]. A 2019 study on the estimated health care costs for fungal diseases in the United States showed a total of 4885 related deaths and an economic burden of over $48 billion USD [332]. The gradual increase in both morbidity and mortality is itself a reflection of the diseases’ high medical costs and impact on worldwide public health. Efforts need to be redoubled and more attention should be paid to basic aspects such as preventive measures and faster and safer lab tests for early diagnosis [333,334]. Above all, there is an urgent need for therapeutic alternatives with antifungal activity and specific mechanisms of action that prove more effective, safer, and cheaper so they are well within reach of the general population. Right now, while medication is prohibitively expensive for low-income patients, phenomena such as human migration will promote disease morbidity and mortality.

Nowadays, scientific and biotechnological advances allow for studies of the genetic interaction between the host and the metagenomic structure of the fungus (mycobiome and microbiome). These studies have revealed, among other things, that fungi have high genomic plasticity and metabolic diversity, factors that have played an essential role in their adaptation to changing environments and their observed evolutionary success over hundreds of years [335,336,337]. There is no doubt that these findings will be very useful for identifying diagnostic markers and designing personalized therapies in the future. Currently, there are some molecular techniques used at the clinical level that allow for diagnostic tests and screening for invasive mycoses [338]; however, their use is also limited given their high cost, which, again, makes them unavailable to the general population.

The development of new cost-effective alternatives to treat invasive mycoses requires a better understanding of the molecular mechanism involved in the antifungal activity of each treatment. Major efforts must be undertaken in the case of gold standards such as AmB. These efforts include (but are not limited to) understanding the physicochemical interactions with cell or lipid membranes, liposomal formulations, and semi-synthetic derivatives.

Polyene antibiotics have been used for more than 60 years to combat protozoa and fungal infections through the induced permeability of the cell membrane. Some have a differentiated action between membranes with cholesterol and ergosterol, with a high degree of effectiveness and generate very low resistance in fungal pathogens [339]. Both Nys and AmB are used to combat fungal pathogens, where Nys is used topically and orally, while AmB is the most effective polyene antibiotic for combating fungal pathogens systemically (e.g., cryptococcal meningitis and invasive zygomycosis) [340,341]. It is also used when there is a lack of response to azole or echinocandin therapeutic treatments in aspergillosis infections, candidiasis, and histoplasmosis [341,342,343,344]. It has an antiparasitic effect against *Trypanosoma cruzi*, *Schistosoma mansoni*, *Echinococcus Multilocularis*, and *Leishmania* spp. [27,344,345,346,347].

Unfortunately, and despite its great efficacy, both AmB and Nys have undesirable collateral toxicity [45,348], the most serious being nephrotoxicity and hematotoxicity [34,49,177], both of which limit their therapeutic use to a short range of specific clinical cases and that can also result in eventual treatment interruption [23,24,25,26]. In spite of these unwanted toxic side effects, AmB is still considered the gold standard for serious invasive fungal infections due to its high antifungal activity, the low appearance of resistant strains, and its low cost compared with other treatments. Its main adverse effects include, in order of importance, nephrotoxicity, hepatotoxicity, and hemotoxicity [349,350,351,352]. Moderate toxic effects such as nausea, vomiting, bloody stool, fever, chills, hypokalemia, hypercalcemia, hypomagnesemia, and increased liver enzymes have also been reported [164,352,353,354,355,356,357,358]. Severe toxicity data indicate the presence of disseminated intravascular coagulation, hypotension, dysrhythmias, renal failure, respiratory failure, and heart failure [359,360].

In order to address said toxicity problems, the pharmaceutical industry has focused for some decades on developing novel formulations that reduce the toxicity profile without affecting therapeutic efficacy [358,359,360], or, as previously stated, novel derivatives. In addition, AmB derivatives have been developed and used in the treatment of human immunodeficiency virus (HIV)-1 to prevent the entry of the virus into P4 cells [31]. Currently, AmB can be found on the market as AmB deoxycholate (Fungizone^®^), AmB lipid complex (Abelcet^®^), liposomal AmB (AmBisome^®^), and AmB in colloidal dispersion (Amphotec^®^). The best-known formulations are AmB lipid complex (ABLC) and liposomal AmB (L-AmB). The last two act as a form of selective drug delivery to the fungal wall [215] while at the same time allowing the drug to remain in circulation and bound to the lipid, reducing tissue distribution and thus avoiding toxicity [361]. Toxicity, however, remains despite these formulation changes. A comparative study to evaluate the safety of the liposomal formulation vs. the AmB lipid complex showed that the administration of 5 mg/Kg of the AmB lipid complex was associated with infusion-related reactions such as fever and chills, although to a lesser degree than with AmB deoxycholate. The use of the AmB lipid complex was also associated with reduced treatment discontinuity given the absence of evident toxicity [362]. Because nephrotoxicity is still present, albeit to a lesser degree, its use in patients with kidney disease remains restricted or requires additional supervision [363,364,365]. The toxic effects at the renal level have already been studied, and we now know that liposomal AmB can induce acute kidney injury by inducing tubular injury and renal vasoconstriction [366]. Tubular injury is the product of intramembranous pore formation or vacuolation of epithelial cells in the distal convoluted tubule [156], whereas renal vascular resistance is increased by activation of the tubuloglomerular feedback mechanism [367].

Which of the two formulations works better in the clinic? We do not know. There are no reports in the literature comparing the efficacy and safety of both formulations given the difficulty of finding two similar populations for this purpose. The clinical status of each patient varies, as do the comorbidities, the pathogenic agent, risk factors, and history of previous therapies, all of which make a comparative clinical study difficult to carry out. However, there have been several attempts using patient populations with specific characteristics. The best option for antifungal treatment will depend on the price of the drug and the patient’s or the health system’s ability to cover this cost. The price of AmB varies, with deoxycholate being the cheapest and most toxic, and liposomal and lipid complex formulations being the most expensive given their production costs. Semi-synthetic derivatives of existing polyenes could prove to be a cost-effective alternative but, up to now, none of the previously presented derivatives are in clinical use. There is a clinical preference for the use of liposomal AmB and there are few studies on the use of AmB lipid complex, probably because the latter is more expensive and has different properties.

### 6.2. Pharmacokinetic Changes

Studies in animals and humans have shown changes in the pharmacokinetic profile of AmB. For example, in a study using rats, the AUC_(0–24 h)_ was 316, 325, and 76 µg/mL; the half-lives for AmB liposomal, lipid complex, and deoxycholate were 9.7, 6.25, and 6.9 h, respectively. Data obtained from patients also indicate that lipid formulations have a higher bioavailability and longer half-lives [368,369,370,371,372,373,374,375,376,377,378]. In a pharmacokinetic study of animals with skin lesions, plasma concentrations were much higher (11-fold) for liposomal AmB than AmB deoxycholate [195]. AmB per se is known to have a high affinity for plasma proteins (>95%), albumin and α1-acid glycoprotein, which causes it to become pharmacologically inactive. Putting it in the bloodstream in encapsulated form prevents its binding to plasma proteins, thereby increasing its mean residence time, plasma concentrations and, consequently, pharmacological activity [379].

Most preclinical studies consistently indicate that liposomal AmB increases drug disposition in organs such as the lung and central nervous system [380,381,382]. Clinical trial results, on the other hand, indicate that liposomal AmB and AmB lipid complexes have better bioavailability, tolerability, and safety compared with AmB deoxycholate [207,208,383,384,385]. Studies conducted in patients have shown the presence of high concentrations of liposomal AmB in the liver, spleen, kidney, thyroid, bone marrow, and lung [386]. A similar tissue distribution was found in patients prescribed AmB deoxycholate, although concentrations were higher in liver, kidney, and spleen, which correlates with the toxicity of the latter formulation [378]. The tissue distribution of AmB has a similar pattern across the different formulations; the difference lies in the amounts found in each organ [387,388].

Fungal infections in the CNS are difficult to treat with AmB deoxycholate because there is some difficulty in crossing the blood–brain barrier (BBB). AmB is a substrate for p-glycoprotein, an efflux transporter found in biological membranes of primary brain capillary endothelial cells, and this places a limit on how long the drug can remain within the cell [389]. Furthermore, Ambisome does not seem to cross the BBB in normal physiological conditions [390], but inflammation and damage to the site secondary to fungal invasion increase the permeability of the BBB [390,391,392]. In a study of a pediatric cohort with hemato-oncological diseases treated with lipid complex AmB, AmB was found in cerebrospinal fluid (CSF) samples and remained for around 48 h in this compartment [387]. Recently, a sensitive method used to quantify lipid complex AmB (1.8 mg/kg) in biological samples from a patient with neuro cryptococcosis yielded important pharmacokinetic data, including that AmB lipid complex becomes widely distributed and was present in CSF (although concentrations were 30 times lower than those found in plasma), again demonstrating that AmB in lipid formulations can cross biological barriers [387,393]. When it comes to urinary excretion, we know AmB is eliminated without metabolizing in the urine and is excreted in very low quantities. In a comparative study, intact AmB was found in the urine of 20.6% of subjects treated with AmB deoxycholate and 4.5% of those treated with liposomal AmB. In turn, it showed in the fecal excretion of 42.5% patients treated with AmB deoxycholate, but only 4% for the subjects treated with liposomal AmB [394,395]. The reduction in the excretion of liposomal AmB may either reflect its longer half-life or mean that the liposomal formulation favors its metabolism and directs its elimination to the kidneys, albeit in the form of metabolites. Several publications show evidence of lipid formulations producing important changes in the absorption, distribution, metabolism, and excretion properties of AmB [214,396,397,398,399,400].

### 6.3. Pharmacodynamic Changes

There are several advantages to the modifications of the pharmacodynamic properties of the lipid formulations. For example, liposomal AmB, in addition to having the same spectrum of activity as AmB deoxycholate, can be used against invasive fungal infections in patients who are refractory or intolerant of conventional AmB. While AmB lipid complex has broadened its spectrum of activity towards filamentous fungi (*Fusarium* spp., phaeohyphomycetes/dematiaceous, fungi/black fungi, *Schizophyllum* and other basidiomycetes, *Scopulariopsis* spp., *Penicillium* spp., *Paecilomyces*), it also covers other yeast fungi (*Saprochaete* spp., *Sporobolomyces* spp., *Trichosporon* spp.) and can be used as empirical therapy for situations where a fungal infection is suspected in patients with febrile neutropenia [401]. Additionally, lipid AmBs have extended their use to other pathogens such as leishmaniasis [402] and mucormycosis [403]. Among some of the other advantages of lipid formulations is the expanded therapeutic index of AmB, its LD_50_ is 5 times higher than AmB deoxycholate, which allows the dose to be increased without increasing its toxic effects [404], thus raising the safety margin. Animal studies have shown that the median lethal dose (LD_50_) was 2 to 3 mg/kg [405] for AmB deoxycholate, 40 mg/kg for liposomal AmB [406], and 175 mg/kg for AmB lipid complex [386]. On the other hand, the standard usage doses for AmB deoxycholate range from 0.25 to 1 mg/Kg, while formulations of liposomal AmB have a 1 to 5 mg/Kg range and lipid complex AmB ranges from 3 to 5 mg/Kg [407]. In the case of the latter, the dose can be further increased or the treatment time extended. Recently, a study conducted in patients with different fungal infections (histoplasmosis, paracoccidioidomycosis, cryptococcal meningitis, and mucocutaneous leishmaniasis) measured trough concentrations of liposomal AmB under different intermittent dosage regimens: at doses of 100 mg/day (4 to 5/week), 50 mg/day (4 to 5/week) and 50 mg/day (1 to 3/week), with an approximate dose of 0.7 to 4 mg/Kg during the initial as well as the consolidation phases of the treatment (7 days). Regardless of the dosage regimen, trough concentrations remained constant in patients, particularly those with cryptococcosis. It has therefore been suggested that intermittent administration regimens of liposomal AmB should be implemented in patients with mycoses [408].

Since their pharmacological profile was modified to improve their bioavailability and safety, lipid formulations have undoubtedly had an impact on the treatment of invasive mycoses. However, studies are still underway to find more about dosage, treatment schemes, combinations with other drugs, and identify new adverse reactions in specific clinical situations. Additionally, effects are still being evaluated in patients in critical conditions or with specific comorbidities.

In spite of the advantages shown by the liposomal formulations, the difference in cost has prevented the complete replacement of AmB deoxycholate formulation. However, the problem is not only cost-effective, as the liposomal formulation poses some problems and seems to be more hepatotoxic [409,410]. Since the advantages of the liposomal formulations reduce toxicity, a derivative presenting increased safety in solution is an important alternative, one that could even be applied in a liposomal formulation.

The clinical application of polyenes is still at the threshold of new developments. Some recent advances for particular applications are described below.

### 6.4. Kidney Damage

Several authors have pointed out that the lipid formulations of AmB have a better safety margin. However, the medical community is still concerned about its suitability for patients with kidney damage, being this organ is AmB’s target. Invasive fungal infections in critically ill patients with acute kidney injury are two comorbidities often observed in the clinic that require prompt treatment to save the patient’s life. Here we present a brief recount of the available evidence of kidney damage associated with the use of liposomal AmB.

Clinical studies indicate that the use of liposomal AmB is empirically advisable, even when the fungal species causing the disease is unknown [411,412]. All cases have resulted in an improvement of the patient’s clinical condition and many of them have achieved recovery. It was recently pointed out that when a daily infusion is administered for 7 days from the onset of acute kidney injury, the patient will exhibit early recovery without alterations in creatinine levels, which indicates there is no deterioration in renal function [413]. Why is liposomal formulation less toxic in the kidney? Probably because it does not concentrate in that organ, and the higher plasma concentrations facilitate the eradication of the pathogenic agent, furthering the patient’s recovery. A recent and retrospective study of 507 patients treated with liposomal AmB (doses of 2–2.5 mg/kg for a period of 7 to 28 days) found certain risk factors associated with the development of acute kidney damage in subjects receiving liposomal AmB. Some of these factors included previous treatment with ACE inhibitors/ARBs or carbapenems, ongoing treatment based on catecholamines or immunosuppressants, or doses ≥ 3.52 mg/kg/day of liposomal AmB. Another additional factor is that the presence of serum potassium levels < 3.5 mEq/L can also lead to severe kidney damage after the administration of liposomal AmB. These findings are of great clinical relevance, since they provide guidelines for the management of patients with invasive fungal infections who present kidney damage. Furthermore, they show that liposomal AmB will only cause kidney damage under certain circumstances [414]. Liposomal formulations for polyene delivery might therefore enable an increased use of these drugs.

### 6.5. Septicemia

Septicemia is a life-threatening bodily response to an infection where the immune system damages different tissues, including the kidney, lung, heart, and nervous system. Septicemia is progressive and can lead to septic shock, which increases the risk of death. A recent study using liposomal AmB in 141 patients with invasive fungal infection and septic shock found that early use of liposomal AmB (>6 mg/kg, for 15 days) was associated with a shorter duration of septic shock and the absence of mortality. That study showed that the timing of liposomal AmB administration was associated with a good prognosis for patients. This is of great clinical importance, especially for the management of those patients with septicemia who are at high risk of developing septic shock due to their immunosuppressed state, other risk factors for poor prognosis, and the pathogenic agent present [415].

### 6.6. Transplants

Fungal infections are very common in transplant patients and are the leading cause of death in immunocompromised patients. For that reason, it is important to monitor them and take timely prophylactic measures or treatment. Very recently, a case of a patient with a liver transplant and refractory invasive candidiasis was published: caspofungin (0.12 mg/L) was administered for 10 days, followed by isavuconazole (0.25 mg/L) for 4 days, then caspofungin (0.12 mg/L) for 4 days and, finally, liposomal AmB (0.25 mg/L) for 12 days, with no response to treatment. A combined therapy based on isavuconazole + liposomal AmB was therefore employed, with a remission of invasive candidiasis 12 days later. This study showed that a combined therapy of liposomal AmB with another antifungal could improve response in cases of persistent mycoses among immunocompromised patients [416]. Prophylactic therapies or extension of liposomal AmB dosage regimens have also been suggested for organ transplant patients [417]. In animals with invasive fungal infections, liposomal AmB has been administered at doses of up to 20 mg/Kg, achieving survival in animals with histoplasmosis as well as neutropenic mice [418,419]. The literature reports human cases where liposomal AmB was used at doses of up to 7.5 mg/kg in adults with transplants and also in children with neutropenia [420]. In a cohort study of 900 patients receiving continuous renal replacement therapy due to acute renal failure, liposomal AmB (0.5 mg/kg) was well tolerated and there was no need to adjust dosage or change the duration of treatment. Patients who underwent hemodialysis and continuous renal replacement therapy had a low incidence of adverse reactions, even if the patients had renal insufficiency. Therefore, liposomal AmB may be indicated in patients who, due to renal dysfunction, require hemodialysis or continuous renal replacement therapy [421].

The use of antimycotics as a prophylactic strategy in patients with Hematopoietic Stem Cell Transplantation (HSCT) is a measure to reduce the presence of invasive mycoses after transplantation. Therefore, when choosing a treatment, it is important to consider the efficacy and safety of the medication. A retrospective study with a cohort of 84 pediatric HSCT patients showed a high incidence of invasive mycoses when liposomal AmB or micafungin were used as prophylactic treatment. Additionally, their use was associated with the presence of nephrotoxicity and hepatotoxicity. This incidence, however, was related to the degree of immunosuppression, type of transplant, or environmental exposure to the pathogen. Therefore, the prophylactic use of liposomal AmB in pediatric patients with CMHP is not recommended because, in addition to not being effective, pediatric patients seem to be more susceptible to toxic effects [422].

Prophylactic therapy is a practice also used in immunocompromised patients scheduled to undergo a transplant. Prophylactic therapies have been using doses of up to 10 mg/kg/week with positive response from patients [423], although the increases in serum creatinine levels suggest impaired renal function. The above findings show that, given the safety of lipid formulations, it is feasible to increase doses or extend treatment span, always monitoring renal function. The prophylactic use of liposomal AmB has been recommended in onco-hematological patients at high risk of fungal infections. While the prophylactic therapy of first choice in these patients is azoles, this therapy is replaced by liposomal AmB when the use of azoles is contraindicated [424].

### 6.7. Anti-Parasitic

For some years now, liposomal AmB has been one of the therapeutic options to treat visceral leishmaniasis in several countries [425]. Unlike what happens with other conditions, there is no established dosage regimen to treat this disease, nor does it follow a characteristic clinical picture. The fact that the therapeutic response varies depending on the species of *Leishmania* or the geographical area makes it difficult to establish a dosage schedule. For example, treatments for leishmaniasis caused by L. donovani in India use up to 10 mg/kg of liposomal AmB, while doses of up to 30 mg/kg are used in Africa [426,427]. For leishmaniasis caused by *L. infactum*, doses of 20 mg/kg are used in America and Europe [428]. It has been reported that, under certain circumstances (HIV infection, transplants, age, presence of other co-infections, etc.), doses of liposomal AmB may reach up to 60 mg/kg [429]. A systematic review of the use of liposomal AmB for the treatment of visceral leishmaniasis has shown evidence not only of its efficacy but, more importantly, of its safety, even in the case of very high doses [430]. In another recent retrospective study, the different AmB formulations were compared for efficacy and safety. Evidence showed liposomal AmB was more effective and less toxic than the other formulations, suggesting it to be the most acceptable treatment for this disease [431]. However, liposomal AmB has not only been employed to treat leishmaniasis of the visceral type; it is also the standard treatment and formulation for mucocutaneous leishmaniasis. We know that liposomal AmB has proved effective in this case because it prevents the macrophage–parasite linkage and inhibits the production of g-INF, which induces macrophage activation. To achieve remission of the disease, the WHO has recommended establishing a regimen based on liposomal AmB at doses of 2–3 mg/kg/day up to a total dose of 40–60 mg/kg to ensure successful results without producing nephrotoxicity. A new treatment in a murine model has shown that a combination of benznidazole and A21 is effective for trypanosomiasis of a very virulent strain [432], offering a possible treatment for Chagas disease.

### 6.8. Mucormycosis

The incidence of mucormycosis has increased significantly in recent years, even in patients without immunodeficiencies. One characteristic of these infections is that they are resistant to most antifungals and their treatment is limited. Liposomal AmB has been effective when administered for prolonged periods of time (4–12 weeks) [433]. Since liposomal AmB can cross the BBB in mycotic infections, it has been proposed as an initial treatment strategy for patients with mucormycosis of the central nervous system at doses of 5–10 mg/kg/day for 28 days. So far, the response has been satisfactory, although monitoring the immune status of the patient is recommended [434]. The combined use of liposomal AmB with other antifungal drugs has reportedly been successful in several clinical trials. A recent retrospective study pointed out that the combination of liposomal AmB together with posaconazole produced a significant synergistic effect for ensuring short-term survival in patients with hematologic malignancy and was more effective than monotherapy [435]. Similar results were seen in a patient with acute lymphoblastic leukemia, although the drug of choice in this case was AmB lipid complex [436]. This evidence shows the potential synergistic interaction that can be obtained with AmB in lipid formulation.

The effectiveness of AmB, regardless of formulation, is clear. As we have seen, the lipid formulations have improved pharmacological profiles and potential interactions, as well as new therapeutic uses. New dosage regimens have been established. Still, lipid formulations are not readily available to all patients suffering from invasive mycoses, which continues to limit the treatment of a large swathe of the global population. The development of new, effective, safe and, above all, low-cost polyene derivatives is therefore urgent. These should be accessible to the general population and help reduce the morbidity and mortality caused by invasive fungal infections.

## 7. Summary and Outlook

Polyenes are small pharmaceutical molecules that produce a great variety of phenomena on lipid membranes and biological cells, and many studies have sought to understanding the mechanism of interaction between molecules and membranes, obtaining valuable information in the process.

Polyene antibiotics remain relevant more than 70 years after they were first introduced into the market. The high antifungal activity and low resistance incidence to polyene treatment are the main reasons why they are still being studied so as to circumvent their substantial host-toxicity. New formulations, such as lipid-based formulations and new semi-synthetic derivatives, bring new life to these drugs while offering therapeutic alternatives for invasive fungal infections and the surge of multiple drug resistant strains such as *Candida auris*. To better design these lipid-based formulations or direct the synthesis of derivatives, we need a solid understanding of polyene selectivity towards fungal cells. In this regard, recent evidence suggests that membrane structure is an important factor. In particular, the evidence suggests that the coexistence of liquid–gel or ordered–disordered phases favor polyene activity at the membrane level. This structure-dependent activity can be exploited to obtain polyenes or polyene formulations with a higher therapeutic index for clinical use.

We expect that future work could be directed to the study and characterization of pore formation in lipid bilayers that present phase segregation and its relation to real fungal or mammalian cell membranes. Another topic that will probably prove quite important in the upcoming years is the immunomodulatory effect of polyenes and their different formulations. As is the case with the pore-forming model, the immunomodulatory phenomenon has yet to be fully understood and is sure to receive a lot of attention.

## Figures and Tables

**Figure 1 membranes-12-00681-f001:**
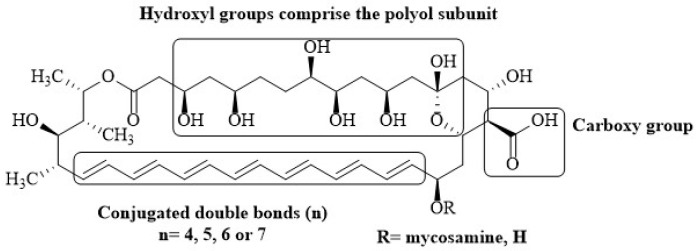
General structure of polyene antibiotics.

**Figure 2 membranes-12-00681-f002:**
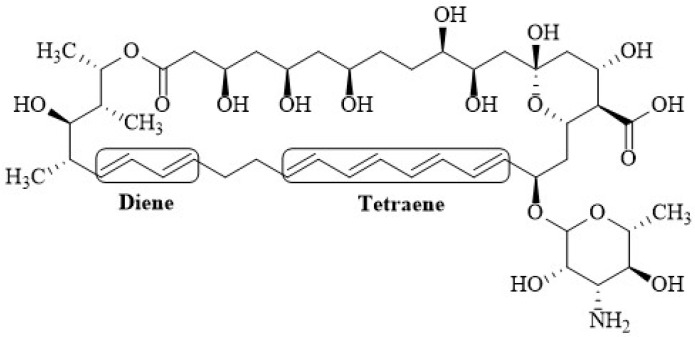
Structure of Nystatin **2**.

**Figure 3 membranes-12-00681-f003:**
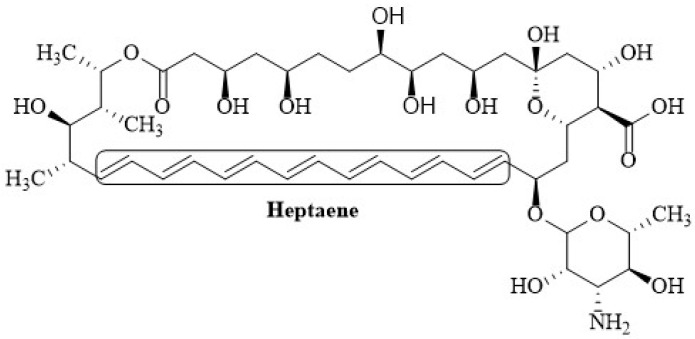
Structure of Amphotericin B **1**.

**Figure 4 membranes-12-00681-f004:**
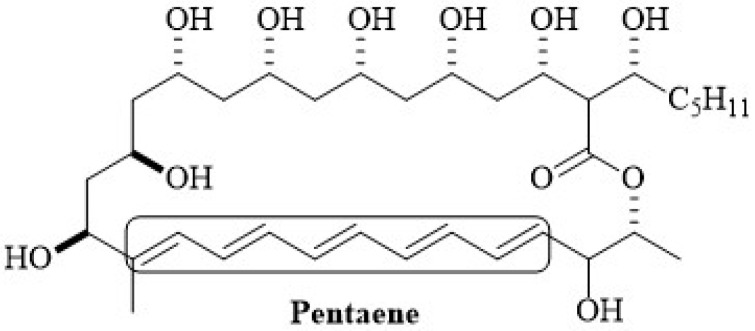
Structure of Filipin III **3**.

**Figure 5 membranes-12-00681-f005:**
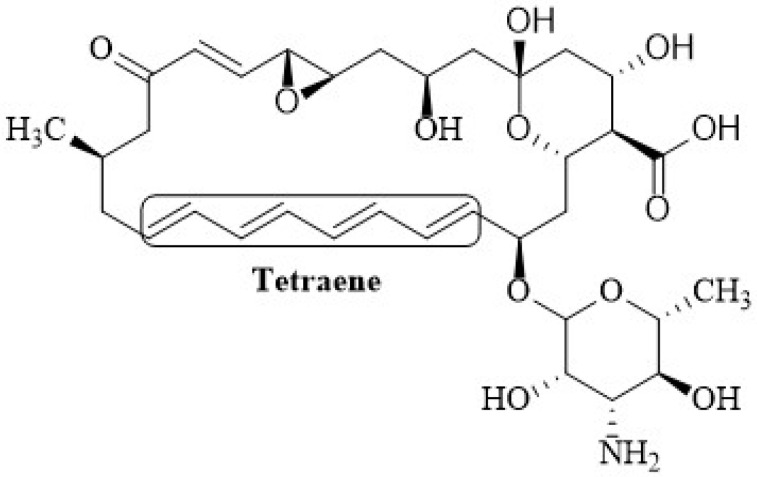
Structure of Natamycin **4**.

**Figure 6 membranes-12-00681-f006:**
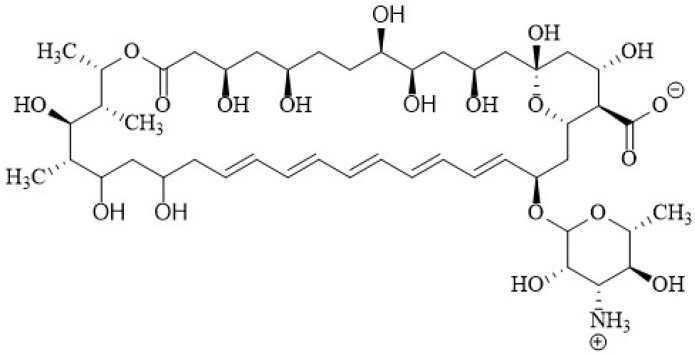
Chemical structure of oxidized amphotericin B form (AmB-ox, n = 5), where n is the number of double bonds in the hydrophobic chain.

**Figure 7 membranes-12-00681-f007:**
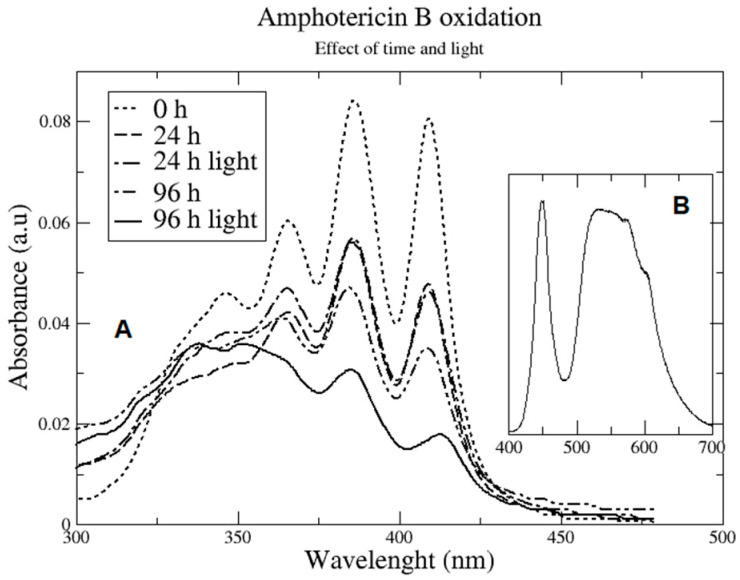
(**A**)—Absorption spectra of Amphotericin B in PBS at pH 7.4 and exposed to air at ambient conditions as a function of time and illumination (2000 luxes). (**B**)—Spectra of the LED lamp used for illumination.

**Figure 8 membranes-12-00681-f008:**
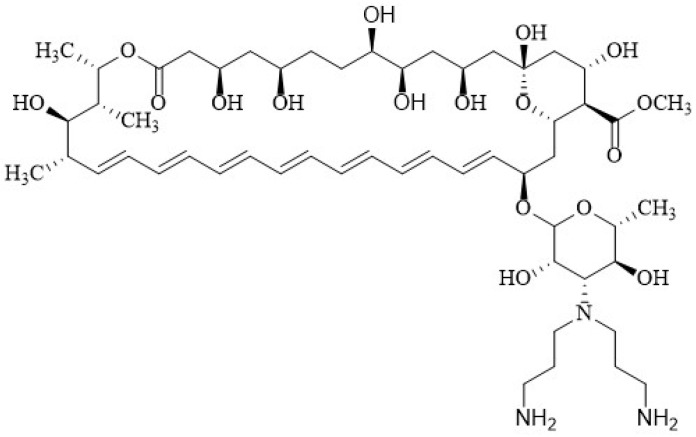
The *N*,*N*-di-(3-aminopropyl) AmB derivative **5**.

**Figure 9 membranes-12-00681-f009:**
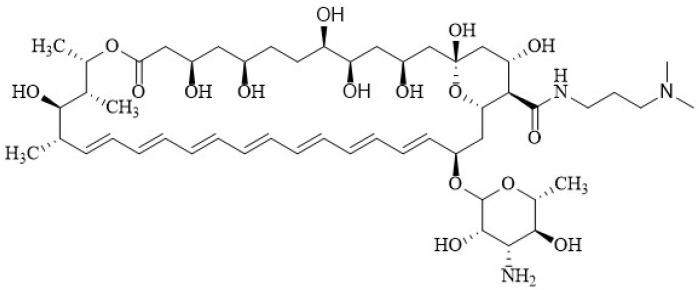
The amide *N*,*N*-dialkyl derivative **6**.

**Figure 10 membranes-12-00681-f010:**
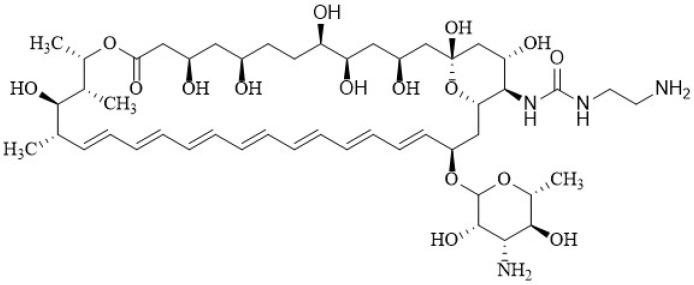
Chemical structure of AmB urea derivative **7**.

**Figure 11 membranes-12-00681-f011:**
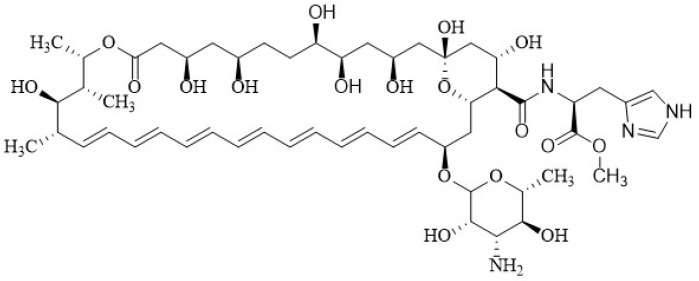
Structure of **A21**, an amphotericin B derivative **8**.

**Figure 12 membranes-12-00681-f012:**
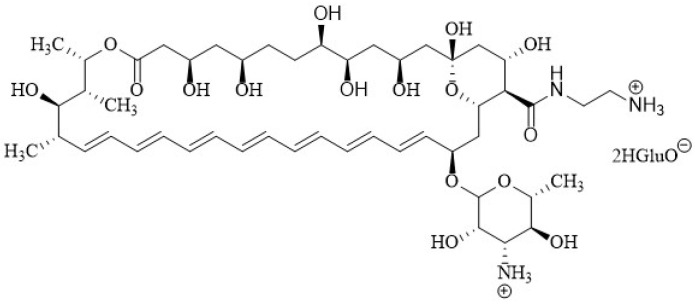
Amphamide salt form with glutamate **9**.

**Figure 13 membranes-12-00681-f013:**
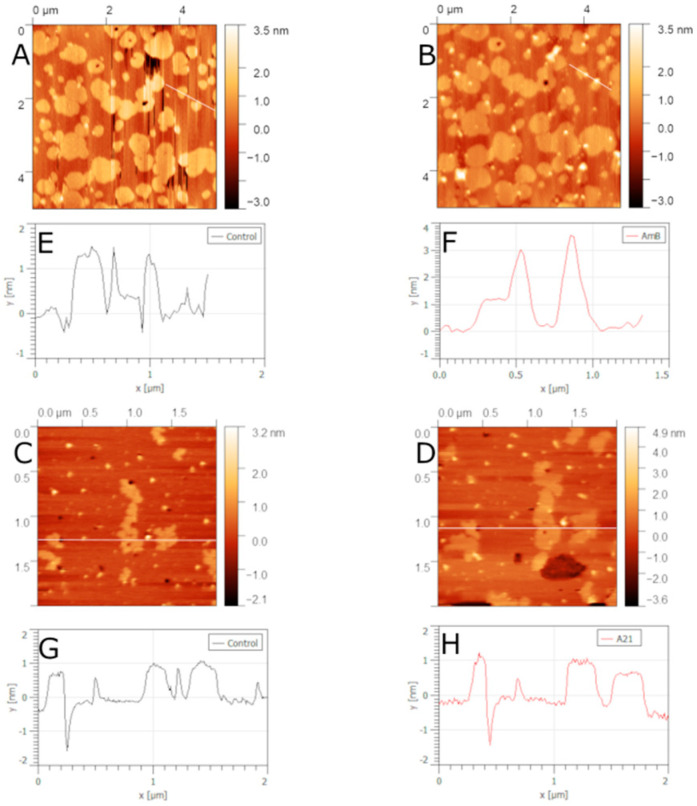
Topographic images of supported lipid bilayers composed of POPC:ESM 1:1 mol:mol + 20 mol% ergosterol in the presence (**B**,**D**) or absence of polyenes (**A**–**C**). Polyenes were added by increments of 0.5 μM until a clear effect was observed. For AmB (**B**), this occurred at 1 μM; for A21 at 2 μM. (**E**–**H**) show height profiles corresponding to the lines drawn on the topographic image.

**Figure 14 membranes-12-00681-f014:**
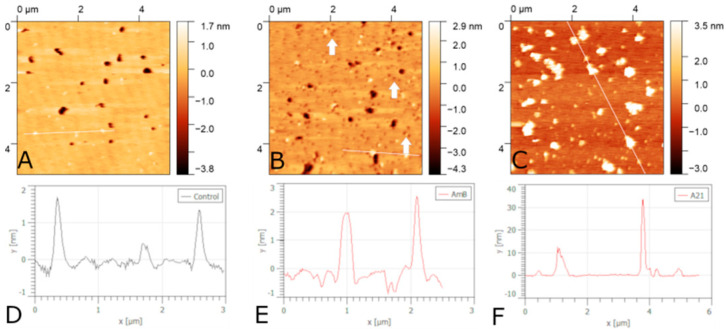
Topographic images of supported lipid bilayers composed of POPC:ESM 1:1 mol:mol + 20 mol% cholesterol in the presence (**B**,**C**) or absence of polyenes (**A**). Polyenes were added by increments of 0.5 μM until a clear effect was observed. For AmB (**B**) this occurred at 2 μM; for A21 at 5 μM. (**D**–**F**) show height profiles corresponding to the lines drawn on the topographic image. White arrows indicate regions where nano-defects or lesions are clearly visible.

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
