# Peer review of "Polyene Antibiotics Physical Chemistry and Their Effect on Lipid Membranes; Impacting Biological Processes and Medical Applications"

_membranes, 2022, doi:10.3390/membranes12070681_

Round 1
Reviewer 1 Report
This is a very thorough and pertinent review on polyene antibiotics. There is, however, a major problem: the English language usage is very poor, starting with the title. A very deep improvement is essential, minor changes will not do.
Also, the paper deals essentially with polyene antibiotics. The word "antibiotics" should be included in the title, and in the Abstract.
Author Response
We have added the word antibiotics to the title and in the abstract.
The manuscript has been rewritten by a professional.
Reviewer 2 Report
In the manuscript, the authors reviewed the polyene physical chemistry and their effect on lipid membranes. The review involves the polyene structure and chemical behavior, interaction mechanisms, polyene host-toxicity, role of membrane structure on the polyene activity, and clinical use of polyenes. This review is in detail and meaningful. Here, I have only two problems to address.
1. The authors are better to cite some new literatures, for examples, the new literatures about the formation of pores in membranes.
2. To better understand, some Figures should be extracted from the cited references and included in this review.
Author Response
The manuscript has been rewritten by a professional.
We made a further search in the literature and add several comments and references.
Unfortunately inclusion of figures copy righted from the literature is not possible for us or the Journal.
Round 2
Reviewer 1 Report
The revised version of this manuscript has taken into account my suggestions for improvement.